# Highly Specific and Rapid Detection of Hepatitis C Virus Using RT-LAMP-Coupled CRISPR–Cas12 Assay

**DOI:** 10.3390/diagnostics12071524

**Published:** 2022-06-23

**Authors:** Nang Kham-Kjing, Nicole Ngo-Giang-Huong, Khajornsak Tragoolpua, Woottichai Khamduang, Sayamon Hongjaisee

**Affiliations:** 1Division of Clinical Microbiology, Department of Medical Technology, Faculty of Associated Medical Sciences, Chiang Mai University, Chiang Mai 50200, Thailand; khamkjing.kk@gmail.com (N.K.-K.); khajornsak.tr@cmu.ac.th (K.T.); 2Maladies Infectieuses et Vecteurs: Écologie, Génétique, Évolution et Contrôle (MIVEGEC), Agropolis University Montpellier, Centre National de la Recherche Scientifique (CNRS), Institut de Recherche Pour le Développement (IRD), 34394 Montpellier, France; nicole.ngo-giang-huong@phpt.org; 3Associated Medical Sciences (AMS)-PHPT Research Collaboration, Chiang Mai 50200, Thailand; 4Infectious Diseases Research Unit, Faculty of Associated Medical Sciences, Chiang Mai University, Chiang Mai 50200, Thailand; 5Research Institute for Health Sciences, Chiang Mai University, Chiang Mai 50200, Thailand

**Keywords:** hepatitis C virus, HCV RNA, chronic hepatitis C, point-of-care testing, RT-LAMP, CRISPR–Cas12, lateral flow-based assay, fluorescence-based assay

## Abstract

Hepatitis C virus (HCV) infection can be cured with pan-genotypic direct-acting antiviral agents. However, identifying individuals with current hepatitis C remains a major challenge, especially in resource-limited settings where access to or availability of molecular tests is still limited. The goal of this study was to develop and validate a molecular assay for the rapid detection of HCV RNA in resource-limited settings. It is based on a combination of reverse transcription loop-mediated isothermal amplification (RT-LAMP) with the clustered regularly interspaced short palindromic repeats–CRISPR-associated protein 12a (CRISPR–Cas12a) cleavage assay that allows the recognition of specific HCV nucleic acid sequences. Amplified products after the cleavage reactions can be visualized on lateral flow strips or measured with a fluorescence detector. When tested on clinical samples from individuals infected with HCV, HIV, or HBV, or from healthy donors, the RT-LAMP-coupled CRISPR–Cas12 assay yielded 96% sensitivity, 100% specificity, and 97% agreement as compared to the reference method (Roche COBAS AmpliPrep/COBAS TaqMan HCV Test). This assay could detect HCV RNA concentrations as low as 10 ng/µL (an estimated 2.38 Log_10_ IU/mL). Therefore, this sensitive and specific assay may represent an affordable and reliable point-of-care test for the identification of individuals with active hepatitis C in low-resource settings.

## 1. Introduction

Hepatitis C virus (HCV) poses a serious global health burden, with an estimated 58 million chronic HCV-infected cases worldwide and about 1.5 million new-infected cases annually [1]. Approximately 60–80% of individuals acquiring HCV develop chronic hepatitis and subsequently severe liver diseases such as cirrhosis and liver cancer [2]. Recently, direct-acting antiviral drugs (DAAs), especially pan-genotypic DAAs, have been shown to be highly effective in curing HCV infection in >95% of treated people [3]. In 2016, the World Health Organization (WHO) proposed eliminating hepatitis C as a public health problem by 2030. Currently, it is estimated that only 21% of those infected with HCV have been diagnosed, and about 62% of those diagnosed have received treatment [1]. A major gap in the cascade of HCV care is the low number of people diagnosed. This is due to the fact that most infected individuals are unaware of their HCV infection, since it can remain asymptomatic for many years. Another reason is the access or availability to diagnosis tests remains limited in many resource-limited settings [2]. Thus, scaling up the screening and diagnosis of hepatitis C is key to achieving the HCV elimination goal. Simple, rapid, accurate, and low-cost HCV diagnostics are crucially needed to identify people who need treatment and to prevent continued virus transmission.

The detection of antibodies to HCV (anti-HCV) identifies only people who had been exposed to HCV. The diagnosis of current HCV infection relies on the detection of HCV RNA, mostly based on real-time reverse transcription polymerase chain reaction (RT-PCR) techniques [4]. However, their use in-clinic is still limited in many low- and middle-income countries (LMIC), since they require specific laboratory infrastructure, well-trained personnel, and high-cost reagents/equipment/maintenance. To overcome these challenges, a simple technique called “loop-mediated isothermal amplification (LAMP)” has been developed [5,6]. This nucleic acid-based amplification method uses four to six primers and isothermal conditions to amplify specifically and effectively the target sequence within an hour [7]. Reverse transcription LAMP (RT-LAMP) is used for the amplification of RNA templates. A major advantage of the LAMP is that it requires a simple equipment providing constant temperature, therefore eliminating the need for thermocycler. Moreover, LAMP products can be visualized by the naked eye through dye indicators or turbidity [6,8]. RT-LAMP assays with colorimetric detection have demonstrated a high sensitivity and specificity to detect HCV in clinical specimens [6,9]. However, this technique has some disadvantages, e.g., nonspecific amplification, cross-contamination, and primer–dimers formation, which may affect the colorimetric reaction and lead to a false positive interpretation [10].

A recent technology called the “clustered regularly interspaced short palindromic repeats–CRISPR associated protein (CRISPR–Cas) system” developed for gene editing and the detection of nucleic acids has been used to diagnose various infections [11,12]. The CRISPR–Cas system includes 2 classes, 6 types, and 19 subtypes [13]. Of these, the type V-A, class 2 CRISPR–Cas system LbCas12a derived from *Lachnospiraceae bacterium* is a unique effector protein with both endoribonuclease and endonuclease activities. It possesses a distinct feature known as the collateral (*trans*-cleavage) activity, a nonspecific single-stranded DNA (ssDNA) cleavage activity following the recognition and cleavage of the specific target [14,15]. Recently, CRISPR–Cas12 combined with the RT-LAMP technique has been extensively used for the detection of RNA viruses in particular [16,17]. The CRISPR–Cas12 system has not yet been proposed for the detection of HCV RNA. Therefore, in this study, we aimed (1) to develop a rapid RT-LAMP-coupled CRISPR–Cas12 assay for the detection of HCV RNA using a lateral flow strip or a fluorescence detector (Figure 1 and Figure 2) to evaluate the performances of the developed assay in terms of specificity and sensitivity and validate its application with clinical samples from HCV-infected individuals and other blood-borne viruses.

## 2. Materials and Methods

### 2.1. Design of CRISPR RNA (crRNA)

The crRNA was designed to specifically detect the 5′-noncoding region (NCR) of all HCV genotypes (Figure 2). To identify the conserved target sequence regions for crRNA binding, HCV sequences were retrieved from the Los Alamos HCV Sequence Database [18] and were aligned using BioEdit software (version 7.2.5). The sequence 5′-/UAAUUUCUACUAAGUGUAGAUGGCGUGCCCCCGCGAGACUGCUA/-3′ was used as crRNA (the spacer sequence is underlined). The crRNA was synthesized by Integrated DNA Technologies, Coralville, IA, USA.

### 2.2. Development of the RT-LAMP-Coupled CRISPR–Cas12 Assays

#### 2.2.1. RT-LAMP Reaction

Viral RNA was extracted using the NucleoSpin RNA Virus kit (Macherey-Nagel, Düren, Germany) according to the manufacturer’s protocol. Extracted RNA was amplified using the RT-LAMP assay previously described, with some modifications [6]. Briefly, a total volume of 25 µL of LAMP reaction composed of 1.6 μM forward inner primer, 1.6 μM reverse inner primer, 0.2 μM forward outer primer, 0.2 μM reverse outer primer, 0.4 μM forward loop primer, 0.4 μM reverse loop primer, 1.4 mM deoxynucleotide (dNTP) mix, 8 mM MgSO4, 8 U *Bst* 2.0 WarmStart DNA Polymerase (New England Biolabs, Ipswich, MA, USA), 7.5 U WarmStart RTx Reverse Transcriptase (New England Biolabs, Ipswich, MA, USA), and 5 µL of RNA template. The reaction was carried out at 65 °C for 60 min and inactivated at 80 °C for 10 min. The LAMP product was then subjected to CRISPR–Cas12 assays.

#### 2.2.2. CRISPR–Cas12 Assay with Lateral Flow-Based Readout

To obtain the optimal intensity of the test band on lateral flow strip, various concentrations of lateral flow cleavage reporters (from 62.5 to 1000 nM) and various incubation times (at 10 min and 30 min) were evaluated. The assay was performed according to the previously published report, with slight modifications [19]. An amount of 5 µL of LAMP product was added to a total volume of 20 µL of CRISPR–Cas12 reaction mixture containing 250 nM LbCas12a (New England Biolabs, Ipswich, MA, USA), 500 nM crRNA, 62.5–1000 nM lateral flow cleavage reporter (5′-/56-FAM/TTATTATT/3Bio/-3′, Integrated DNA Technologies, Coralville, IA, USA), and 1X NEBuffer 2.1. The mixture was then incubated at 37 °C for 10 or 30 min, after which, 100 µL of HybriDetect assay buffer was added into the mixture. A lateral flow strip (Milenia^®^ HybriDetect, Milenia Biotec, Giessen, Germany) was then immersed into the reaction tube. The result was interpreted as negative if there was a single band on the Control line and as positive if there were two bands on both Control and Test lines, or even a single band on the Test line (Figure 1 and Appendix A). To determine the optimal readout time, the strip was read at 2, 3, 4, and 5 min, respectively, after immersing the strip into the reaction tube.

#### 2.2.3. CRISPR–Cas12 Assay with Fluorescence-Based Readout

The CRISPR–Cas12 assay for fluorescence-based readout was performed as previously described by Broughton et al., with a slight modification, i.e., the elimination of the preincubation step for the RNA–protein complex formation [16]. Reaction mixture of 20 µL was prepared from 40 nM LbCas12a, 40 nM crRNA, and 100 nM fluorescence reporter molecule (5′-/56-FAM/TTATTATT/3BHQ_1/-3′, Integrated DNA Technologies, Coralville, IA, USA). Then, 2 µL of LAMP product was added to the mixture. Green fluorescence signal was monitored every 30 s for 30 min at 37 °C using FAM channel of the Applied Biosystems 7500 Fast instrument (Thermo Fisher Scientific Inc., Waltham, MA, USA).

### 2.3. Evaluation with Clinical Samples

To evaluate the efficacy of HCV RNA detection in clinical samples, the results obtained from the developed RT-LAMP-coupled CRISPR–Cas12 assay were compared with a commercial real-time RT-PCR method (COBAS AmpliPrep/COBAS TaqMan HCV Test; Roche Molecular Systems, Pleasanton, CA, USA). Plasma samples were collected from 100 HCV-infected individuals who had HCV genotyping (using in-house direct sequencing) and routine HCV viral load measurements (using COBAS AmpliPrep/COBAS TaqMan HCV Test) at the Clinical Microbiology Laboratory Service Unit, Division of Clinical Microbiology, Department of Medical Technology, Faculty of Associated Medical Sciences (AMS), Chiang Mai University (CMU). All samples used in this study were anonymized with a study code.

The specificity of the RT-LAMP-coupled CRISPR–Cas12 assay was assessed with plasma samples from 10 HBV-infected and 10 HIV-infected subjects who routinely tested for viral loads at the Faculty of AMS, CMU, and from 10 healthy blood donors at the Blood Bank Section, Maharaj Nakorn Chiang Mai Hospital. All samples were kept at −70 °C until testing. Plasma samples were extracted using the NucleoSpin RNA Virus kit (Macherey-Nagel, Düren, Germany), according to the manufacturer’s protocol. Extracted RNA underwent RT-LAMP and CRISPR–Cas12 assays. Results were interpreted with both lateral flow-based and fluorescence-based readouts.

To evaluate the limit of detection (LoD), 1 µL of five tenfold serial dilutions (ranging from 100 to 0.01 ng/µL) of extracted RNA of HCV genotypes 1, 3, and 6 were subjected to RT-LAMP as described above with duplicate at each dilution. LAMP product of each dilution was then subjected to CRISPR–Cas12 assays with both lateral flow-based and fluorescence-based readouts.

### 2.4. Ethical and Biosafety Statements

This study was ethically approved by the Faculty of Associated Medical Sciences Ethic Committee (AMSEC-64EM-007) and authorized by the Institutional Biosafety Committee of the Research Institute for Health Sciences, CMU (CMUIBC0364001).

### 2.5. Statistical Analysis

The variables with binary results between the RT-LAMP-coupled CRISPR–Cas12 and the reference assays were used to calculate the specificity, sensitivity, positive predictive value (PPV), and negative predictive value (NPV). The 95% confidence intervals (95% CI) for the proportion were calculated using the free online calculator (GraphPad: https://www.graphpad.com/quickcalcs/ConfInterval1.cfm, accessed on 8 March 2022).

## 3. Results

### 3.1. Optimization of the CRISPR–Cas12 Assay Conditions with Lateral Flow-Based Readout for HCV RNA Detection

With the positive control (PC), at 10 min of incubation, the band intensities of the Test and Control lines did not vary across the range of reporter concentrations used, 62.5 to 1000 nM (Figure 3a). At 30 min of incubation, the Test bands were intense at 500 and 1000 nM, but the Control bands were either faint or absent, indicating a complete cleavage of the reporter by Cas12 (Figure 3b). With the no template control (NTC), the intensities of the Test bands were inversely related to the reporter concentrations either at 10 or 30 min of incubation (Figure 3c,d). Therefore, 1000 nM lateral flow reporters at 30 min incubation time were chosen as optimal conditions for further lateral flow-based readout.

The optimal readout time was also assessed after 2 to 5 min of strip immersion. The Test bands of PC were clearly observed during that time range, while the Test band intensities of the NTC slightly increased according to the readout time (Figure 3e). Thus, the assay readout time was subsequently set at 2 min.

### 3.2. Clinical Evaluation of RT-LAMP-Coupled CRISPR–Cas12 Assay for HCV RNA Detection

Of 100 plasma samples with known HCV viral loads, 93 samples were positive by RT-LAMP-coupled CRISPR–Cas12 assay with both readouts. The results of the first 30 representative for HCV samples are shown in Figure 4a,b, while the rest showed similar patterns. Seven samples with inconsistent results were retested; of these, three samples became positive with both readouts (Table 1). The other four samples which remained negative had HCV viral loads of 3.96, 4.06, 4.85, and 6.58 Log_10_ IU/mL (Table 2). On confirmation of these four samples with agarose gel electrophoresis, no band was observed.

For specificity testing, 30 non-HCV templates (i.e., 10 HBV-, 10 HIV-infected individuals, and 10 healthy blood donors) were used. All samples showed no Test band on the lateral flow strip (Figure 4c) and no signal of fluorescence. Therefore, after retesting, the developed RT-LAMP-coupled CRISPR–Cas12 assay either lateral flow-based or fluorescence-based readouts demonstrated 96% sensitivity (95% CI, 90–99%), 100% specificity (95% CI, 87–100%), and 97% agreement (95% CI, 92–99%). The PPV and NPV were 100% (95% CI, 95–100%) and 88% (95% CI, 73–96%), respectively.

The LoD of the RT-LAMP-coupled CRISPR–Cas12 assay with lateral flow-based readout was determined using various RNA concentrations of HCV genotypes 1, 3, and 6. Positive results were observed from 10 ng/µL (an estimated 2.38 Log_10_ IU/mL) for all three genotypes tested (Figure 5). LoD results with fluorescence-based readout were in accordance with those from the lateral flow-based readout (data was not shown).

## 4. Discussion

We developed and validated an assay for rapid detection of HCV RNA based on an RT-LAMP-coupled CRISPR–Cas12 system that can be deployed at the point-of-care to identify people with current HCV infection. To our knowledge, this is the first report demonstrating HCV RNA detection based on the combination of the RT-LAMP with CRISPR–Cas12 assay with two types of readout, lateral flow strip or fluorescence measurement.

This developed assay presents several advantages over the existing platforms based on the RT-LAMP or real-time RT-PCR technologies. First, this assay only required a simple equipment, e.g., water bath or heat block, able to provide a constant temperature. Indeed, RT-LAMP reaction is done at 65 °C and the CRISPR–Cas12 assay at 37 °C. Second, the combination of the RT-LAMP with CRISPR–Cas12 module provided the assay with higher specificity than with a LAMP assay solely, with which false amplification through nonspecific nucleic acid amplification can occur, leading to false positive results [20]. The advantage of using the LbCas12a-crRNA complex is that it can discriminate the nucleotide mismatches, especially on the seed region [14,21]. Moreover, this assay can detect HCV RNA within approximately 1 h with fluorescence measurement or within approximately 1 h 30 min through lateral flow-based readout after extraction step. Thus, this diagnosis approach may be a conceptual or suitable for on-site HCV RNA detection as compared to the real-time RT-PCR. The lower LoD of the RT-LAMP-coupled CRISPR–Cas12 assay in this study was 10 ng/µL for genotype 1, 3, and 6, while that of the RT-LAMP using dye staining were 10, 10, and 100 ng/µL, respectively [6]. The combination of RT-LAMP with the CRISPR–Cas assay is able to increase the sensitivity of the assay by signal amplification via the *trans*-cleavage activity, which can be reversible [14].

The developed the RT-LAMP-coupled CRISPR–Cas12 assay which showed 96% sensitivity and 100% specificity when evaluated with clinical samples. Of the 100 samples collected from the HCV-infected individuals, four samples remained negative after retesting. Of these four, three had HCV viral load below 5 log_10_ IU/mL (3.96, 4.06, 4.85) and one had HCV viral load of 6.58 log_10_ IU/mL. This discrepancy may result from the degradation of the RNA due to the long storage duration or multiple freeze and thaw cycles. Assessment of other blood-borne viruses, such as HIV and HBV, showed no cross-reactivity. Our results are consistent with those of a study that reported no cross-reaction of the CRISPR–Cas12a assay to detect HBV [22]. Therefore, the coupling of the RT-LAMP with the CRISPR–Cas12 assay offered a highly specific and robust detection of HCV RNA.

As noted, we still observed a very faint band on the Test line of the negative samples. In contrast to the healthy donors, a very strong Test band was observed in most of patients with low HCV viremia, suggesting that the intensity of positivity was not dependent on patient viral load but the cleavage activity of the CRISRP–Cas (Appendix A). This faint Test band can be explained by the high dose hook effect, a phenomenon typically associated with the immunoassay, and which also occurs in lateral flow assays. To avoid this faint Test band, it is necessary to determine the optimal amount of the reporter in the lateral flow assay. Another option is to pipet the reaction mixture into the lateral flow strip instead of immersing the strip into the reaction tube. Finally, it may be necessary to develop a lateral flow strip that is specific for the CRISPR–Cas system.

The performances of this assay indicate it can be transferred to clinical settings at the point-of-care, particularly with a lateral flow-based readout. Indeed, this system is portable, disposable, easy to use, and suitable for large-scale detection. Moreover, it is a low-cost assay (less than 20 USD) with a shorter turnaround time compared to the real-time RT-PCR. In addition, the CRISPR–Cas12 assay with the fluorescence-based readout can be exploited for shortening the turnaround time, since the fluorescence signal can be detected after 1 min of the CRISPR–Cas12 reaction. The increase in the fluorescence intensity did not depend on the viral load, but rather on the cleavage efficiency of the Cas12–crRNA complex and the cleavage time. At the point-of-care, fluorescence signals can be measured with a commercial portable/handheld fluorimeter or an in-house smartphone-based fluorescence detector with machine learning-driven software [23,24]. However, new technology based on the fluorescence biosensor, oligonucleotide sensor, colorimetric biosensor, and electrochemical biosensors have been recently developed, and may represent promising approaches for the diagnosis of infectious diseases [25]. For example, the nucleocapsid of SARS CoV-2 could be rapidly detected using an aptasensor based on a microelectrode array chip [26]. Moreover, biosensors have been developed to improve food safety, e.g., metal–organic frameworks (MOFs)-based fluorescence and electrochemical biosensors that can detect very low levels of food-borne pathogens or any food contaminants due to their high affinity for immobilizing analyte-targeted probes (DNA, aptamers, and antibodies) [27].

Our study has some limitations. First, not all HCV genotypes were included in the clinical evaluation of the assay. Only samples with HCV genotype 1, 3, 6, and one sample of genotype 4 were included as these genotypes are predominant in our setting. However, our assay is still of interest since genotypes 1 and 3 are the most prevalent globally and genotype 6 is commonly found in Southeast Asia [28]. In fact, we designed the primer sets and the guided crRNA based on sequences of all genotypes suggesting that our assay can be used to detect other HCV genotypes. Second, no internal RNA control was included. RT-LAMP amplification is known to be a robust technique, as it is insensitive to polymerase inhibitors that may be present in clinical samples [29]. To verify a possible inhibition in a reaction tube, a two-tube approach can be applied, one for the target gene and another one for the internal control (spiked HCV RNA), or a multiplex CRISPR–Cas assay using different substrates can be carried out. Anyway, a further real-world evaluation study is still needed to ensure the feasibility of the developed assay in a routine use.

## 5. Conclusions

We validated a highly specific, rapid, and accurate RT-LAMP-coupled CRISPR–Cas12 assay for HCV RNA detection. This assay is easy to use and has a low cost, and thus can be rapidly implemented in resource-limited settings to identify people who need to be treated for their HCV infection. This assay can contribute to the ultimate goal of hepatitis C elimination from the public by 2030.

## Figures and Tables

**Figure 1 diagnostics-12-01524-f001:**
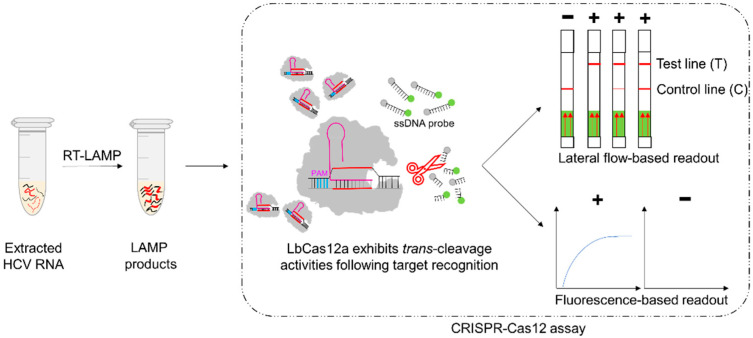
Overall workflow for RT-LAMP-coupled CRISPR–Cas12 assay. Extracted HCV RNA was amplified using RT-LAMP assay and the LAMP products was detected by CRISPR–Cas12 assay. Results were interpreted through lateral flow-based or fluorescence-based readouts. T, Test line; C, Control line; PAM, protospacer adjacent motif.

**Figure 2 diagnostics-12-01524-f002:**
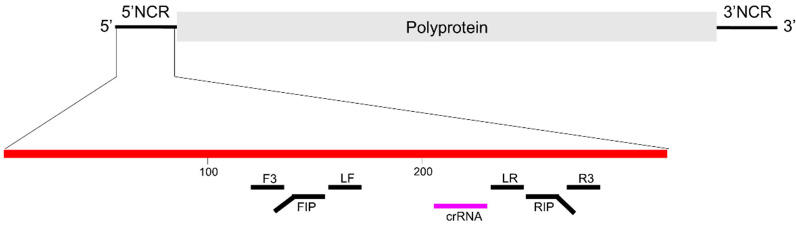
Genome map with the binding positions of RT-LAMP primers and crRNA. Red line indicates the 5′ NCR of HCV genome, the black lines indicate the LAMP primers, and the purple line represents the crRNA. NCR, noncoding region; F3, forward outer primer; R3, reverse outer primer; FIP, forward inner primer; RIP, reverse inner primer; LF, loop forward primer; LR, loop reverse primer; crRNA, CRISPR RNA.

**Figure 3 diagnostics-12-01524-f003:**
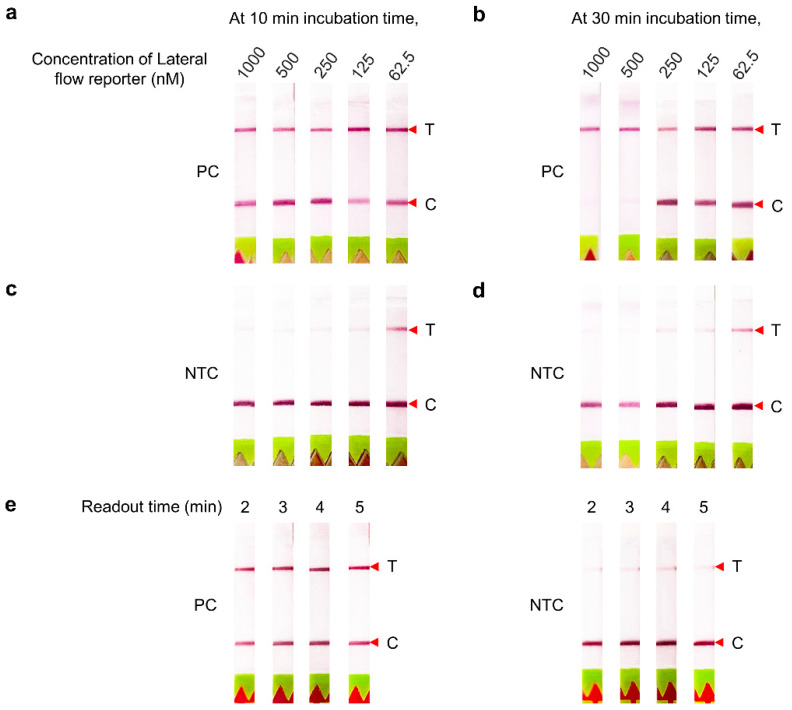
Optimization of CRISPR–Cas12 assay for HCV RNA detection using lateral flow-based readout. (**a**) Optimization using an HCV positive sample/control with various concentrations of lateral flow reporter, i.e., 1000, 500, 250, 125, and 62.5 nM at 37 °C for 10 min or (**b**) 30 min. (**c**) Optimization using NTC with various concentrations of lateral flow reporter at 10 min or (**d**) 30 min. (**e**) Results at different readout time of an HCV positive sample/control and NTC. T, Test line; C, Control line; PC, positive control; NTC, no template control.

**Figure 4 diagnostics-12-01524-f004:**
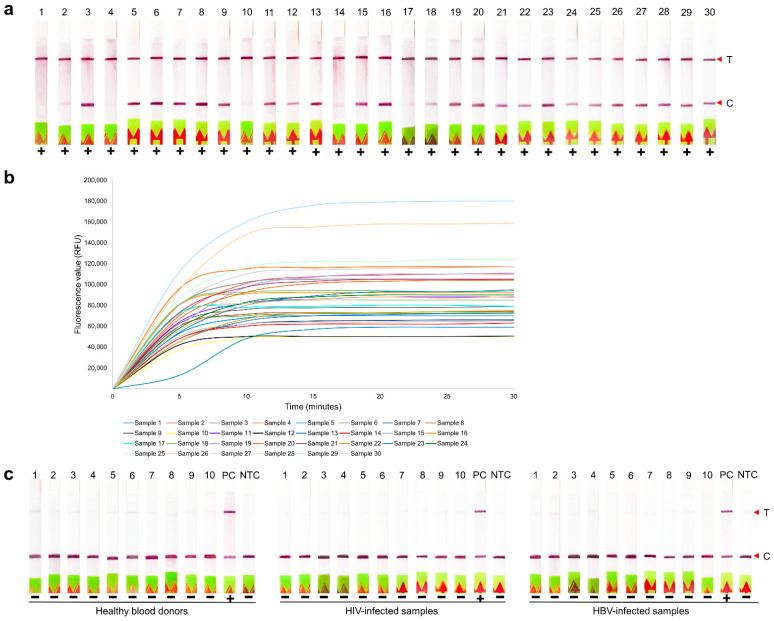
Evaluation of the developed RT-LAMP-coupled CRISPR–Cas12 assay. (**a**) RT-LAMP-coupled CRISPR–Cas12 assay for HCV RNA detection using lateral flow-based readout and (**b**) fluorescence-based readout among the first 30 representative clinical samples from HCV patients while the rest showed similar patterns; (**c**) RT-LAMP-coupled CRISPR–Cas12 assay for HCV RNA detection using lateral flow-based readout among 10 healthy blood donors, 10 HIV- and 10 HBV-infected samples. T, Test line; C, Control line; PC, positive control; NTC, no template control.

**Figure 5 diagnostics-12-01524-f005:**
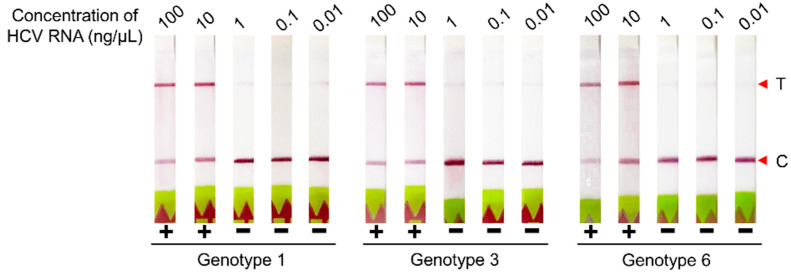
Determination of LoD of the RT-LAMP-coupled CRISPR–Cas12 assay. LoD was performed with HCV genotype 1, 3, and 6 at various input concentrations of HCV RNA: 100, 10, 1, 0.1, and 0.01 ng/µL. T, Test line; C, Control line.

**Table 1 diagnostics-12-01524-t001:** Clinical performance of the RT-LAMP-coupled CRISPR–Cas12 assay.

Assays		RT-LAMP-Coupled CRISPR–Cas12 Assay							
		(1st Round)				(2nd Round)			
		Lateral Flow-Based Readout		Fluorescence-Based Readout		Lateral Flow-Based Readout		Fluorescence-Based Readout	
		Positive (%)	Negative (%)	Positive (%)	Negative (%)	Positive (%)	Negative (%)	Positive (%)	Negative (%)
Real-time RT-PCR	Detectable (N = 100 ^a^)	93 (93)	7 (7)	93 (93)	7 (7)	96 (96)	4 (4)	96 (96)	4 (4)
	Undetectable (N = 30 ^b^)	0 (0)	30 (100)	0 (0)	30 (100)	ND	ND	ND	ND

^a^ 100 HCV positive samples; ^b^ 30 negative samples including 10 HIV-, 10 HBV-infected, and 10 healthy blood donor samples; N, total number of samples tested; ND, not done.

**Table 2 diagnostics-12-01524-t002:** Evaluation of the RT-LAMP-coupled CRISPR–Cas12 assay according to HCV genotypes and viral loads.

HCV Viral Load (Log_10_ IU/mL)	All Samples	Genotype 1	Genotype 3	Genotype 4	Genotype 6
	*n*/*N* (%)	*n*/*N* (%)	*n*/*N* (%)	*n*/*N* (%)	*n*/*N* (%)
7.01–8.00	15/15 (100)	7/7 (100)	5/5 (100)	NA	3/3 (100)
6.01–7.00	34/35 (97)	15/16 (94)	12/12 (100)	NA	7/7 (100)
5.01–6.00	24/24 (100)	10/10 (100)	10/10 (100)	1/1 (100)	3/3 (100)
4.01–5.00	21/23 (91)	9/10 (90)	10/11 (91)	NA	2/2 (100)
3.01–4.00	2/3 (67)	2/2 (100)	0/1 (0)	NA	NA
Total	96/100 (96)	43/45 (96)	37/39 (95)	1/1 (100)	15/15 (100)

*n*, total number of positive; *N*, total number of samples tested; NA, not available.

## Data Availability

Not applicable.

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
