# Peer review of "Highly Specific and Rapid Detection of Hepatitis C Virus Using RT-LAMP-Coupled CRISPR–Cas12 Assay"

_diagnostics, 2022, doi:10.3390/diagnostics12071524_

Round 1

Reviewer 1 Report

The authors have presented a specific and a rapid diagnostic POC for HCV. It will be good and be used in remote areas or in resource-poor countries. I have very few and minor comments and of the view that once these are addressed, the manuscript is acceptable for publication. 

Minor Comments:

- In the abstract, "DNA amplified products obtained from cleavage reaction" is misleading. It should be "after the cleavage reaction" 

- In the abstract: LoD of HCV RNA was mentioned in ng/ul. The IU/ml value should also be mentioned in parenthesis for more clarity.  

- Materials Methods: Please mention and underline to sequence for tracer and spacer regions. If there is an appropriate citation, it should be cited. 

- RT-LAMP rxn; 5ul of RNA- Please mention how many nanograms.

- CRISPR-Cas12 assay: What was the wavelength of the florescence signal (FAM or HEX etc). 

Though, clinical samples infected with HCV can be detected with this new technique. Concerning point is that some healthy donors are also showing faint bands in the test window (T). Keeping in mind the false positivity, how would you differentiate between a healthy donor and a very low HCV viremic patient?

Authors should comment on that or it can be improved by doing plasma dilutions mimicking the high to low viral load viremic patients and then extracted RNA be used for lateral flow strips (not the RNA dilutions). 

Overall, the manuscript is well-written and will provide a new POC testing for HCV.  

Reviewer 2 Report

In this work, the authors developed a rapid RT-LAMP-coupled CRISPR-Cas assay for the detection of HCV RNA using a lateral flow strip or a fluorescence detector to evaluate the performances of the developed assay and validate its application with clinical specimens from HCV-infected individuals and other blood-borne viruses. The results are interesting and the method is novel. The introduction and discussion need to be revised for clearance of background. PLEASE revise your paper based on the following issues:

-abstract need to rewritten to give the goal of the study

- Some abbreviations were used without explanation

- Properly check for subscripts and superscripts throughout the manuscript.

-it suggests using these interesting articles in the biosensing field to compare with CRISPR-Cas assay: https://doi.org/10.1021/acs.analchem.1c04296, https://doi.org/10.3390/bios12050314, https://doi.org/10.1016/j.tifs.2021.10.024, https://doi.org/10.1016/j.ijbiomac.2022.02.082

- The language of the paper needs to be improved thoroughly

- Conclusion should be rewritten to present the final goal of your review. While writing the conclusion, the author should remember, that the main conclusions of the study may be presented in a short Conclusions section, which may stand alone or form a subsection of a Discussion or Results and Discussion section. The Conclusion should not be a summary but should illustrate the advances and claims of innovative aspects of the research work done.

-recently Bioinformatic gained lots of interest in the field of medicine and especially CRISPR. it can be helpful for the readers to obtain an insight into this new branch of science. I think https://doi.org/10.1111/aos.14928 and https://doi.org/10.3389/fphar.2021.642900 studies are interesting ones to highlight bioinformatic importance.

Round 2

Reviewer 2 Report

I asked the authors to discuss some new refs in the field of biosensors. the suggested refs can help readers to gain a wider insight about currently used biosensors in the field of medicine.
